# General Purpose Artificial Intelligence Systems as Group Agents

**Matija Franklin UCL, Causal Cognition Lab London, UK**
matija.franklin@ucl.ac.uk

## Abstract

This paper advocates for General Purpose Artificial Intelligence Systems to be viewed as group agents. This view emphasizes their shared agency characteristics and allows for the assignment of collective responsibility, while still recognizing individual accountability when necessary. This perspective could streamline AI regulation, promote responsible use, and ensure accountability.

## 1 Introduction

Companies, markets, and social networks often metaphorically get compared to Artificial Intelligence (AI) or even Artificial Super Intelligence (ASI), where the collective intelligence of group members exceeds the ability of one individual group member. This paper argues that regulation and law surrounding General Purpose Artificial Intelligence Systems (GPAIS) could benefit from treating GPAIS as a group agent. GPAIS has been defined as "An AI system that can accomplish or be adapted to accomplish a range of distinct tasks, including some for which it was not intentionally and specifically trained (Gutierrez et al., 2022)."

Entities that classify as agents have a *representational state* that describes how things are in an environment, a *motivational state* that represents how the agent requires things to be in the environment, and finally, it has to have the capacity to process its representational and motivational states in order to intervene in the environment whenever there isn't a match between that environment and the agent's motivation (List & Pettit, 2011). An AI can be considered an agent given its representational state - AI learns representations in its environment, motivational state - AI systems have reward functions (Ashton, 2022), and its ability to intervene - they use their internal representations of the environment to determine the best actions to take to achieve those objectives (Russell, 2010).

A Group agent is a group that exhibits the three core features of agency (List & Pettit, 2006). The members of a group agent can play similar roles or each have distinctive tasks. Further, there must be a basis for thinking of a group agent as the same entity even when individual members of this agent change. Group agents give us a better understanding of the social world as people perceive and talk about groups as agents, which in turn influences their behavior (List & Pettit, 2011). Once a collective entity is recognized as an agent, one can interact with it, criticize it, and make demands on it in a manner not possible with non-agentic systems.

## 2 GPAIS as a Group Agents

Previous research has argued that sole AI agents and group agents have parallels in that both involve non-human entities that qualify as intentional agents, capable of making high-stakes decisions and performing actions on their own (List, 2021). List (2021) argues that "group agents can be viewed as special cases of AI systems, where the "hardware" supporting their artificial intelligence is social rather than electronic." I argue that it is useful to think of GPAIS as a group agent.

An entity made up of multiple individuals can be considered a group agent if it satisfies several key criteria (Pettit & Schweikard, 2006). Firstly, the members must work together to establish common objectives and devise a plan for identifying additional goals in the future. Secondly, the group must collaborate to establish a set of principles to guide their actions in support of these objectives and also establish a process for refining these principles as needed. Finally, the group must determine who will be responsible for carrying out actions toward achieving the goals, whether it be the entire

group, individual members, specific individuals, or agents hired by the group. These criteria result in the emergence of collective intentionality that allows groups to act as a single agent (List & Pettit, 2011). This occurs when individuals all intend to together promote a given shared goal, each intends to make their individual contribution, and are interdependent in that they form intentions because they believe other members have these intentions too.

GPAIS meet these requirements in a number of ways (Bosch et al., 2021; Thórisson, 2007). First, GPAIS can coordinate to set goals and procedures for achieving those goals. Second, GPAIS can make decisions and take actions based on a set of rules or algorithms that are designed to support their goals. This can include using data, past experiences, and other inputs to make decisions and take actions that support the goals. Third, GPAIS can act autonomously, without the need for explicit direction from human operators. An example of this is Auto-GPT - an agent designed to proactively generates and adjusts its own objectives to fulfill larger tasks without the explicit need for continuous human input (Gravitas, 2023).

## 3 IMPLICATIONS OF VIEWING GPAIS AS GROUP AGENTS

People's intuitions, when an AI agent does something wrong, is to allocate blame to a broader group of agents involved in developing and operating that system: a user, company, developer, data provider, dataset, etc (Franklin et al., 2023; 2022; Hidalgo et al., 2021). A similar response occurs when a group agent does something wrong. When a corporation's activities result in environmental damage individuals will often identify several individuals to whom the blame can be attributed. However, in both the case of an AI system or a company, it is often not appropriate to target one or several individuals as the responsible agent. Otherwise, the user or developer of an AI will often be the agents that are held responsible. As argued by List (2021) "Perhaps all humans involved—the system's operators, developers, owners, and regulators—were sufficiently diligent and acted conscientiously, and nonetheless the AI system caused a harm." It is important to note that treating GPAIS as a group agent would still allow us to punish one or several individuals where this is appropriate.

Treating GPAIS as group agents gives us an existing framework for addressing issues pertaining to AI regulation. GPAIS can be regulated as a group agent that can be held responsible for its actions (List, 2021). Such an approach would not be far removed from what is currently socially normative. Experimental jurisprudence research has also found that people in a hypothetical jury were willing to attribute intent to AI ( 50% of the participants) (Ashton et al., 2022), and are more broadly willing to attribute blame to AI ( 80% of the participants) (Franklin et al., 2023; 2022).

If GPAIS can be held responsible, they - or their legal representatives - can be regulated and held accountable for breaking the law. This includes but is not limited to:

- **Monetary penalties**: financial punishments as a consequence for non-compliance with laws, regulations, or other stipulated rules.
- **Continuous monitoring**: auditing of AI systems can help identify and address problems early on, reducing the likelihood of harm.
- **Transparency**: providing information and documentation about operations and decision-making processes.
- **Public relations penalties**: suffering reputational damage when engaged in unethical or illegal practices, which result in a loss of public trust and a decline in use.
- **Revocation of licenses**: GPAIS that violate regulations can have their licenses revoked, which may result in the ability to legally operate in certain areas.

## 4 CONCLUSION

This paper advocates for a paradigm shift in the regulatory framework and legal considerations for GPAIS by viewing them as group agents. This view emphasizes their shared agency characteristics and allows for the assignment of collective responsibility, while still recognizing individual accountability when necessary. This perspective could not only streamline AI regulation in line with societal norms and legal precedents, but also promote responsible use and ensure accountability in the emerging AI-dominant landscape.

URM STATEMENT

Author MF meets the URM criteria of ICLR 2023 Tiny Papers Track.

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
