# OpenReview forum: "General Purpose Artificial Intelligence Systems as Group Agents"
_ICLR.cc/2023/TinyPapers — Submitted to Tiny Papers @ ICLR 2023_

### Official Review · Reviewer_kged · 2023-03-19

**Confidence:** 3

**Summary Of Contributions:**

The authors of the paper suggest that considering General Purpose Artificial Intelligence Systems (GPAIS) as group agents could benefit the regulations and laws surrounding these systems. The authors establish a connection between entities, agents, and group agents, and apply this framework to AI systems. The authors main argument is that - GPAIS are group agents; and further discuss the potential impact of this approach.

**Rating:**

Great Start (GS): a submission which meets some of the reviewing criteria but has room for improvement

**Strengths And Weaknesses:**

- **Strengths**
    - The authors explain a conceptual framework that creates a linkage between entities and agents by utilizing a combination of representational states, motivation states, and actions. They then extend this connection to encompass the relationship between individual agents and group agents, and proceed to apply this framework to AI systems.
    - The authors provide an interesting view point and attempt to provide an explanation for holding GPAIS accountable for any violations of the law.
- **Weakness**
    - The paper would benefit by an analysis of the potential risk of overgeneralization. Group agents are designed to operate in specific contexts and may not be applicable in all situations. GPAIS, on the other hand, are designed to be general-purpose and adaptable to different contexts and tasks. By modeling GPAIS as group agents, we risk recognizing their distinct strengths and weaknesses.
    - In the third section of the paper, the author states that -” However, in both the case of an AI system or a company, it is often not appropriate to target one or several individuals as the responsible agent. Otherwise, the user or developer of an AI will often be the agents that are held responsible.”  In the Conclusion section, the author states - “the benefit of doing so is that when a GPAIS does something wrong, its developers and users do not need to be blamed when this is not appropriate”.  While this approach is a step towards ensuring that developers and operators of AI systems are not unfairly held legally responsible for their actions, there are potential drawbacks that need to be considered. One drawback is that this approach could lead to a situation where individuals and organizations are not held accountable for harmful or unethical actions. This statement may oversimplify the complex decision-making processes of AI systems and companies. While it is true that the actions of these entities are often the result of interactions among many different individuals and subsystems, it is still important to understand the specific roles and responsibilities of these actors in order to hold them accountable for their actions. Therefore, a nuanced discussion is required to determine what is considered "appropriate" in holding GPAIS responsible. The author should expand on this work and explore the potential limitations and drawbacks of holding GPAIS responsible. Transparency and explainability of AI systems could be an avenue for further exploration.
    - The concept of responsibility also implies that an agent is capable of making intentional decisions and being aware of the consequences of their actions. It is unclear if a future AI system could possess such characteristics and that of moral agency.

**Suggested Changes:**

I would suggest that the authors provide more detailed descriptions and explanations of key concepts such as collective intentionality and group agents to facilitate a better understanding of their work, especially in the context of this paper.

For instance, in the Introduction section, the authors mentions - “Group agents are a group that exhibits the three core features of agency” as outlined in List & Pettit (2006). It would be helpful if the authors could specify which of the four conditions of agency defined by List & Pettit (2006) they are referring to in order to provide greater clarity.

---

### Official Review · Reviewer_YQZT · 2023-04-04

**Confidence:** 3

**Summary Of Contributions:**

This paper proposes the philosophical thought that treating General Purpose Artificial Intelligence Systems as group agent can be beneficial in formulating laws and regulations surrounding their use and an existing framework can be used for laws related to these bodies

**Rating:**

Great Start (GS): a submission which meets some of the reviewing criteria but has room for improvement

**Strengths And Weaknesses:**

The paper is well-written, proposes the idea of treating GPAIS as group agents, and defines how they qualify to be group agents. The paper then goes on to describe what will be the implication of viewing GPAIS as group agents. The paper concludes that "The benefit of doing so is that when a GPAIS does something wrong, its developers and users do not need to be blamed when this is not appropriate." is slightly flawed as this might result in companies or organisations developing GPAIS and not being withheld for its wrongdoing even if the company/developer was intentionally using it with wrong intentions. Overall, I like the intention of the paper and the way it is written but I might not completely agree with the proposal and it will help in the further development of GPAIS.

**Suggested Changes:**

I would like to see these views backed with some statistics with survey respondents belonging to the varied backgrounds and different stakeholders to fully understand if they make sense. A logical backing might not be sufficient in issues which affect society in broader sense.

---

### Author Response · Authors · 2023-06-01
**I wish to opt-in for archival**

I wish to opt-in for archival

---

### Comment · Area_Chair_5BLD · 2023-06-02
**Updated Meta Review**

This work meets the threshold for archival, contains the URM statement, and is deanonymized.

The paper has been improved a lot after the revision. However, please check the references in the main body. There is a missing reference in the last sentence of the second paragraph in Section 3. Also, make sure the \citet{} and \citep{} are used correctly in the ICLR template. In most cases in the paper, the references should be cited by \citep{}.

---

> ### Author Response · Authors · 2023-06-05
> **Reply**
>
> Thank you for pointing out the issue with the references. I have fixed the issue with the missing reference and the reference formatting and uploaded the revised version.

---

### Meta-Review · Area_Chair_5BLD · 2023-04-08

**Recommendation:** Invite to revise
**Confidence:** 4

**Metareview:**

This paper suggests that considering General Purpose Artificial Intelligence Systems (GPAIS) as group agents could benefit the regulations and laws surrounding these systems. The authors establish a connection between entities, agents, and group agents, and apply this framework to AI systems. The reviewers agree with the interesting view point and attempt to provide an explanation for holding GPAIS accountable for any violations of the law. However, the key concern is that the proposal in the paper may oversimplify the complex decision-making processes of AI systems and companies and lead to a situation where individuals and organizations are not held accountable for harmful or unethical actions.




**Summary:**

 This paper proposes the philosophical thought that treating General Purpose Artificial Intelligence Systems as group agent can be beneficial in formulating laws and regulations surrounding their use and an existing framework can be used for laws related to these bodies. The proposed idea is interesting, however, both of reviewers do not completely agree with the proposal and raise concerns about the risk of overgeneralization.

**Reason For Not Giving A Higher Recommendation:**

Though the intention of this work is good, there are several flaws in the proposal that need more in-depth discussions, i.e., (i) a lack of statistics; (ii) potential risk of overgeneralization; (iii) potential risk where individuals and organizations are not held accountable for harmful or unethical actions.

**Reason For Not Giving A Lower Recommendation:**

N/A

---

### Decision · Program_Chairs · 2023-04-09

Revision accepted; invite to archive